# Music Therapy: A Noninvasive Treatment to Reduce Anxiety and Pain of Colorectal Cancer Patients—A Systemic Literature Review

**DOI:** 10.3390/medicina59030482

**Published:** 2023-03-01

**Authors:** Evan Huang, Jeffrey Huang

**Affiliations:** 1Carrollwood Day School, Tampa, FL 33613, USA; 2Department of Anesthesiology, H. Lee Moffitt Cancer Center, Tampa, FL 33613, USA; 3Department of Oncological Science, University of South Florida, Tampa, FL 33613, USA

**Keywords:** music therapy, colorectal cancer, depression, pain, quality of life

## Abstract

*Background and Objectives:* Music interventions have been used for patients with cancer to meet their psychological, physical, social, and spiritual needs. This review identified the efficacy of music therapy among adult patients with colorectal cancer (CRC). *Materials and Methods:* We searched the PubMed/MEDLINE, CINAHL, and Cochrane Library databases. Only randomized controlled studies reported in English of patients with CRC were included. Two reviewers independently extracted data on patients and intervention measurements. The main outcomes included pain, anxiety, quality of life, mood, nausea, vomiting, vital signs. *Results:* A total of 147 articles were identified from the search. A total of 10 studies were included in the review. Nine out of the ten studies (90%) showed statistically and clinically significant improvements across the outcome variables. Only one study (10%) found no significant positive effect from music therapy in any of the measured outcomes. Among the seven studies measuring pain as an outcome, four studies (57%) demonstrated that music therapy reduced pain. Three studies (75%) showed that MT reduced anxiety. *Conclusions:* This systemic review indicates that music therapy might help reduce pain and anxiety for cancer patients, including those with colorectal cancer, who are receiving treatment in palliative care, inpatient care and outpatient care settings.

## 1. Introduction

Cancer is a major public health problem worldwide and the second leading cause of death in the United States. A total of 1,918,030 new cancer cases and 609,360 cancer deaths were projected to occur in the United States in 2022 [1]. Colorectal cancer (CRC) is the third most common cause of cancer-related death and is the second most common cancer diagnosed in women and the third most in men in the US [2]. In 2022, an estimated 151, 030 new cases of colorectal cancer were expected to be diagnosed [1]. Surgical resection is the core standard-of-care treatment for colon cancer; in addition, patients can also receive fluorouracil-based adjuvant chemotherapy, which can effectively reduce the risk of tumor reappearance and improve overall survival [3,4,5,6,7], as well as decrease the risk of death by 10% to 15% for patients with stage 3 colon cancer [8].

Colorectal cancer can reduce quality of life through the direct and indirect consequences of the disease. On a physical level, these patients have to suffer possible long-term side effects of oncological care (e.g., abdominal pain, fatigue, change in bowel habits, and pain) [2]. On a psychological level, the diagnosis and cancer treatment often increase negative emotions and impair physical and mental well-being [9].

A focus on patients’ mental well-being has started to emerge, as an estimated 35% to 45% of patients with cancer experience some level of distress during diagnosis or treatment [10,11]. Clinical trials have shown that patients face psychological distress starting from the preoperative phase of cancer treatment [12,13]. Significant psychological distress, including anxiety, depression, post-traumatic stress symptoms, fear of cancer recurrence, pain, fatigue, and sleep disturbances [14], impairs quality of life (QOL).

Emotional management plays an important role in quality of life and health management [15]. Emotional management includes patients’ ability to reduce or avoid negative emotions and to understand their own emotional journey in depth to permit the cultivation of new personal resources [15]. Emotional management is especially important for cancer patients. Patients with a history of tumors usually experience fear of recurrence [16,17], which could create pathological anxiety and depression, and may reduce their motivation to continue managing their own health [15]. Healthcare providers should consider each patient’s ability to understand and manage emotions to improve the patient’s physical and mental well-being. 

There are many activities that appear to help improve well-being for cancer patients. These activities include participation in personally enriching activities (e.g., painting, writing, artmaking, and music), challenging activities, such as dragon boating, motorcycle riding, reading, or embracing social relationships [18].

Music therapy (MT) has been used in different medical fields to meet patients’ psychological, physical, social, and spiritual needs. Music has been used to decrease anxiety before or during surgical procedures [19,20], decrease stress during treatment [21,22], reduce nausea and vomiting [23,24,25], decrease pain [26], and improve QOL [27,28,29]. There are two types of MT. Active music therapy requires the patient’s participation in the production of music (e.g., by singing or playing an instrument), whereas receptive music therapy guides the patient in listening to live or recorded music. Music therapy specifically can use music as a way of non-pharmaceutical noninvasive therapy to enhance the emotional well-being of cancer patients, which may help improve cancer patients’ perioperative treatment outcomes.

A study showed that 27.3% of patients with CRC had used complementary medicine [30]. The most used complementary medicine among patients with CRC was mind–body medicine [30], which includes music therapy. Due to the high prevalence and high mortality rates of colorectal cancer and the high utilization of complementary medicine (including MT) in CRC patients, a systematic literature review is needed to summarize the current literature on the application of music therapy as an adjunct noninvasive CRC therapy. Our primary goal is to demonstrate the current literature about music therapy as an adjunct noninvasive treatment for CRC patients. 

## 2. Methods

### 2.1. Search Strategy

For this literature review, we sourced studies regarding music therapy as an adjunct treatment for cancer, as music therapy relates to pain, anxiety, depression. This search was performed using PubMed/MEDLINE (1966 to August 2022), CINAHL (1982 to August 2022), and Cochrane Library databases (up to August 2022) for all articles meeting the search criteria. The literature search was conducted in August 2022.

The search terms used in this review were “Gastrointestinal Neoplasms” OR (“Rectal Neoplasms”) OR (“gastrointestinal cancer”) OR (“rectal cancer”) OR (“Colorectal Neoplasms”) OR (“colorectal cancer”) OR (“cancer patients”) AND (Singing OR choir OR drumming OR music OR “Music Therapy”) AND (“Palliative Medicine”) OR “Palliative Care”) OR (“Pain Management”). Reference lists of all eligible articles were cross-checked for other relevant studies. 

### 2.2. Inclusion Criteria

Inclusion criteria included (a) the study methods must be randomized controlled trials or controlled trials; (b) music therapy must be the interventional treatment; (c) participants must include patients with CRC or gastrointestinal cancer; (d) outcomes must measure pain, anxiety, depression, distress, or QOL; and (e) the article must be reported in English. 

### 2.3. Study Selection and Data Extraction

The results from the selected articles were evaluated by the authors individually (E.H., J. H). Duplicated articles were removed first. Any studies that failed to provide significant relevant data, despite meeting the criteria, were excluded as well. Then, titles and abstracts were screened to exclude irrelevant articles. The full text of selected articles was read and evaluated to examine methodological study quality. Only the studies that met the inclusion criteria were included, and the data were extracted. The primary outcomes included pain, depression, anxiety, distress, quality of life, heart rate (HR), blood pressure (BP), respiratory rate. 

All reviews were performed blind. Discrepancies were resolved through discussions between investigators (EH, JH). Two independent researchers (EH, JH) assessed the quality of included studies using the Cochrane risk-of-bias tool for randomized trials. This decision was made because all studies included in this systematic review were randomized controlled trials (RCTs). Only studies marked as “Low risk of bias” were included in this study.

The articles screened are shown in Figure 1. Data were recorded with the following categories: study, year of publication, country, type of study design, patients, type of cancer, percentage of colorectal cancer/gastrointestinal cancer, MT type and treatment methods, measurements, main results.

This study was conducted by following the Preferred Reporting Items for Systematic Reviews and Meta-Analyses (PRISMA).

## 3. Results

The initial search identified 147 articles (PubMed: 44, Cochrane: 28, and CINAHL: 75). After the initial screening of titles and abstracts, 103 were excluded. A total of 44 articles’ full texts were reviewed and evaluated; 10 studies were included in this review (Figure 1).

Nine out of the ten studies (90%) showed statistically and clinically significant improvements across the outcome variables (Table 1). Only one study (10%) found no significant positive effect from music therapy in any of the measured outcomes. Among the seven studies measuring pain as an outcome, four studies (57%) demonstrated that music therapy reduced pain. Four studies measured anxiety; three studies (75%) showed that MT reduced anxiety.

### 3.1. Demographic Characteristics of Participants

In the 10 studies included in this review, all studies indicated the source of the patients’ cancers [31,32,33,34,35,36,37,38,39,40]. All studies included patients with gastrointestinal cancer, and two studies did not indicate the percentage of GI cancer patients [36,37]. 

Four studies included participants receiving palliative care [34,35,36,37]. Three studies were conducted when participants were hospitalized for in-patient care [31,32,33]. One study included patients receiving chemotherapy [38]. Two studies included cancer patients who had outpatient medical procedures, such as biopsy or vascular port placement [39,40].

All studies were prospective randomized clinical trial studies; these had a control group of either a different relaxation method, such as mindfulness intervention, or a standard-of-care procedure, such as a no-music therapy group. 

### 3.2. Types of Music Intervention 

The length of music played ranged from 10 min to half an hour per session. Many studies had at least two sessions of music therapy [34,35,36,37,39,40]. One study included music played daily [37]. It is noteworthy that some studies that occurred during palliative care and hospice had a wide range of session numbers, as the time from patient enrollment to death ranged widely [37].

The types of music intervention varied greatly among studies, though listening to soft music was used widely [31,33,34,35,36,37,38]. A form of music therapy intervention called the Song of Life, also known as a biographically meaningful song, was used in one study [35]. Two studies used multiple types of musical forms through either active or receptive music therapy [37,38]. In active music therapy, the patients participated in creating the music by singing or playing an instrument, whereas in receptive music therapy, the patients only listened to the music. 

### 3.3. Forms of Measurement

Measurement systems and techniques varied widely across all 10 studies. 

#### 3.3.1. Qualitative Measurement 

Pain was measured via the Pain Numeric Rating Scale in six studies [31,33,34,35,36,39]. Four studies measured changes in anxiety through the anxiety scale [33,36,39,40]. One study measured depression through a depression inventory [31]. One study measured changes in QOL by using quality of life questionnaires [37].

#### 3.3.2. Quantitative Measurement 

One study measured changes in heart rate or blood pressure [33]; one study specifically measured changes in nausea or vomiting [38]; one study measured overall survival [37]; and one study measured changes in pupillary size or respiratory rate [33]. 

### 3.4. Outcomes (Anxiety, Depression, Pain, QOL)

A multitude of outcomes or results were measured among the 10 studies examined, with the most common being anxiety, depression, pain, distress, or QOL. Because of the distinct differences and perspectives that each study examined, the outcome of music therapy among patients with GI/colon cancer was divided into two settings: during hospice or palliative care and during treatment. Each study will be thoroughly examined and reviewed within its respective categories. 

#### 3.4.1. During Hospice and Palliative Care

Both hospice and palliative care treat patients with serious illnesses who require specialized medical care. There is a high risk of mortality and numerous symptoms, and treatments are given with the overall goal of improving the patient’s condition. Hospice is a type of palliative care for patients with a prognosis of 6 months or fewer. Hospice care focuses on end-of-life care that seeks to maximize time at home for patients near death and seeks to avoid hospitalization. 

Huang et al. [34] studied cancer patients who were hospitalized in palliative care, oncology, respiratory, and gastrointestinal floors. Their study setting was mixed palliative care and in-patient care. They proved that music therapy intervention was very helpful by showing their study’s effect sizes, d = 0.64, sensation, and 0.70, distress. There was significantly less post-intervention pain in the music versus the control group, *p* < 0.001. Thirty minutes of music therapy created a half reduction in pain score in 42% of the music therapy group compared to 8% of the controls. They also demonstrated that the type of music (Taiwanese music or American music) had no impact on the results.

Warth et al. [35] showed there was a significant reduction in the NCCN distress score in the music therapy group (an effect size of −0.51 (−0.86, −0.16)). The NCCN distress score recorded initially as 5 decreased to 3.5 near the end of music therapy. The music therapy group patients also reported higher spiritual well-being (an effect size of 0.52 (0.21, 0.84) and ego-integrity (an effect size of 0.72 (0.30, 1.13)). However, they found no differences in in-patients’ functional capacity or their psychological QOL from the control group receiving standard-of-care treatment. 

Düzgün et al. [36] showed that music therapy decreased mean pain scores over time (*p* < 0.001), decreased anxiety (*p* < 0.001), and improved comfort levels (*p* < 0.001). However, Düzgün et al. did not report the effect size.

Hilliard [37] found that music therapy conferred significant differences in overall QOL. There are three subscales in the overall QOL: functional, psychophysiological, and social/spiritual well-being. The authors did further analysis and found that there were no differences in functional well-being (F (l, 78) = 2.034; *p* > 0.05), social, or spiritual well-being (F (l, 78) = 1.278; *p* > 0.05) between the music therapy group and the control group. There was a difference in psychophysical well-being (F (l, 78) = 3.995; *p* < 0.05). The authors also showed that music therapy had no impact on the patient’s length of life. 

#### 3.4.2. During Cancer Treatment

Hunter et al. [38] reviewed 474 participants (6% of whom had gastrointestinal cancer) and examined the effect that music therapy and relaxing music had on managing anticipatory nausea and vomiting during chemotherapy. Significantly fewer participants experienced nausea at the midpoint of chemotherapy, but no difference was found in anticipatory nausea at the endpoint of therapy. The authors did not report the effect size. Additionally, no difference was found in the prevalence of anticipatory nausea or vomiting, at any time point, and quality of life.

Fernando et al. [33] and Arruda et al. [31] examined the effects of music therapy on pain, depression, and anxiety. Both studies found that values of pain, anxiety, and depression were significantly diminished. Aurruda et al. [31] used the Visual Analog Scale (VAS) and the Beck Depression Inventory (BDI) to measure pain and depression. They found that music therapy significantly decreased pain (*p* < 0.001) and depression (*p* = 0.004) (confidence interval = 95%) in patients with cancer who were hospitalized. Additionally, Fernando et al. [33] used the Wong–Baker Faces Pain Scale to measure pain and anxiety. They reported significant reductions in pain (*p* = 0.007, 95% CI: 3.716 to −1.534), anxiety (*p* = 0.0022, 95% CI: −2.465 to −0.6184), respiratory rate, and pupillary size in the music therapy group. They found no differences in pulse rate, systolic blood pressure, and diastolic blood pressure between the music group and the control group. However, this study was a crossover study. 

Bieligmeyer et al. [32] used BMQ (a self-rating tool) to assess the current state of mood by measuring the total sum and various subscales, including inner balance, vitality, vigilance, and social extroversion. They found music therapy slightly increases balance, vitality, vigilance, and satisfaction, as well as significant increases in body warmth, present mood, and overall satisfaction. They did not report the effect size. They found no significant changes in pain levels (*p* = 0.21).

Clark et al. [40] measured the Hospital Anxiety and Depression Scale (HAD), the Distress Numeric Rating Scale, the Profile of Mood States (POMS) Fatigue, and the Pain Numeric Rating Scale. They demonstrated anxiety (*p* < 0.05) and distress (*p* < 0.01) were significantly lower for patients in the music therapy group than the control group. They also found that there were no significant differences between group scores in pain, depression, and fatigue. They did not report the effect size.

Kwekkeboom et al. [39] studied cancer patients undergoing outpatient medical procedures in an outpatient oncology clinic. They measured pain intensity using a numeric rating scale and anxiety using the Speilberger State-Trait Anxiety Inventory–state portion (STAI-s). Pain and post-procedure anxiety experienced by patients in the music therapy group did not differ from those in the control group. The pain score in the music group was X = 2.33 (SD = 0.37), and in the control group, it was X = 1.47 (SD = 0.40). The post-procedure anxiety score was X = 33.45 (SD = 1.77) in the music group and X = 30.59 (SD = 1.93) in the control group. 

## 4. Discussion 

This systemic review demonstrates that music therapy reduced pain, anxiety, and stress and improved QOL among cancer patients, which includes CRC. In this study, nine out of the ten studies (90%) showed statistically and clinically significant improvements across the outcome variables. Only one study (10%) found no significant positive effect from music therapy in any of the measured outcomes. Among the seven studies measuring pain as an outcome, four studies (57%) demonstrated that music therapy reduced pain. Among the four studies measuring anxiety as an outcome, three studies (75%) showed that MT reduced anxiety. The results are consistent with previous systemic reviews and meta-analyses [29,41,42,43]. This review only includes RCTs with participants who had CRC or gastrointestinal cancer. The other systemic reviews [29,41,42,43] include all types of cancer patients. 

Colorectal cancer comprises nearly 10% of all annually diagnosed cancers and cancer-related deaths worldwide [2]. The number of colorectal cancer cases worldwide is expected to increase to 2.5 million new cases in 2035 [2]. CRC is the fourth most frequently diagnosed cancer and the second leading cause of cancer death in the United States. Most of these cases are in adults who are 50 years or older [44]. The median onset age is 67 years. The 5-year relative survival rate has reached almost 65% in high-income countries, such as Australia, Canada, the United States, and several European countries, but remains less than 50% in low-income countries [45,46].

Surgery resection is the cornerstone of curative colon cancer treatment [2]. Laparoscopy has become the standard technique for colon cancer treatment in many countries worldwide [2]. Systemic therapy with adjuvant therapy fluoropyrimidine-based chemotherapy improves survival [47]. The addition of oxaliplatin to fluoropyrimidine (fluorouracil or capecitabine) has become the new standard of care [47]. Surgery with neoadjuvant radiotherapy therapy reduces the rate of local recurrence and toxic effects. 

Colorectal cancer patients carry physical and psychological burdens that affect their quality of life. These sufferings do not end with successful physical care but continue even after treatment. They have to fight negative emotions related to the diagnosis (such as emotional distress and anxiety) and possible long-term side effects of surgery, radiation therapy, and chemotherapy on their health (e.g., fatigue, abdominal pain, a change in bowel functions, and the risk of infections) [2,15].

Colorectal cancer patients can develop significant distress, such as depression and anxiety, after receiving a diagnosis of cancer, which can cause significant levels of distress. However, this is even more prevalent in patients with CRC than other cancer patients because of the possibility of undergoing colon surgery or disruptions in gastrointestinal function [48]. In fact, a study found that 35% to 71% of patients who were preoperative for colorectal resection had symptoms of depression or anxiety [49]. Even more pressing is the positive correlation found between patients with clinical depression and the overall mortality risk, as clinically depressed patients with CRC have a 28% excess mortality risk compared with nondepressed patients receiving similar treatment [50].

It is important to help colorectal cancer patients overcome negative emotions related to the diagnosis, manage their emotions, and improve their health and well-being. Psychological interventions can help cancer patients improve their understanding of emotions, thoughts, and sensations to decrease emotional distress and negative behaviors [51]. After interventions, they demonstrate general satisfaction with their psychological well-being.

As new techniques are developed to help with therapy for patients with cancer, the option of music therapy should not be disregarded. The American Music Therapy Association has defined music therapy as “the clinical and evidence-based use of music interventions to accomplish individualized goals within a therapeutic relationship by a credentialed professional who has completed an approved music therapy program” [52]. Music therapy has been shown to act as an emotional stimulus, activating different parts of the brain [53]. Music effect may affect stress responses from the endocrine system (corticotropin-releasing hormone, corticotropin, and cortisol) and the autonomic nervous system (the release of norepinephrine from central and peripheral sympathetic nerve terminals and the release of epinephrine from the adrenal medulla) [54]. Music therapy has effects on anxiety, pain, fatigue, and QOL in people with cancer and reduces respiratory rate, blood pressure, and heart rate [29]. Our study shows that music therapy reduced pain, anxiety, and stress and improved QOL among colorectal cancer patients.

The results of this study can add other evidence to help clinicians understand the power of music to help their patients in clinical settings. Background music can be used in hospital waiting areas and wards. Music therapists can also help design individual treatment programs for each patient. Music therapy can be applied from an overnight stay to returning regularly to the hospital for recurring therapies. Cancer patients need to visit or stay in hospitals for repeated cancer treatments (chemotherapy, radiation therapy, surgeries, etc.). Therefore, it is important to implement music therapy in hospital settings to improve patient outcomes.

Music therapy can be used as a psychological intervention to help patients to promote personal strength, interrupt the cycle of distress, and improve well-being. However, psychological interventions often require patient engagement and commitment. Patient motivation can facilitate better adherence, retention, and the outcome of therapy [55]. Intrinsic motivation is associated with greater results and long-term benefits [56]. Therefore, patient engagement also plays an important role in achieving these outcomes.

On the basis of the results of this review study, the primary conclusion is that music therapy can reduce pain and anxiety and improve QOL for cancer patients. However, the outcomes were not consistent among all the studies. Conflicting results were identified in different studies. It is difficult to have a generalized conclusion for all cancer patients. This study emphasizes the potential for continuing research to better understand the use of music therapy for patients with cancer.

This study includes some limitations. There is no research examining the use of music therapy among CRC patients specifically. The number of patients with CRC in the included studies was very small, so meta-analysis cannot be applied. The literature search was limited to PubMed/MEDLINE, CINAHL, and Cochrane Library, and it is likely that other eligible studies were not included. There was high heterogeneity in the studies (such as patient population, study setting, outcome measurement methods, music therapy types, and control group selection). 

Further randomized controlled trials should be established not only to generate stronger evidence but also to determine the most effective form of music therapy to understand the true effects of music on patients with CRC. The effectiveness of music therapy in alleviating pain may differ depending on the cancer stage, and further investigation is needed to understand the positive effects of music therapy on cancer patients at different cancer stages. Additionally, more prospective randomized controlled multicenter studies are needed.

## 5. Conclusions

In conclusion, this systematic literature review and qualitative analysis supported the efficacy of MT for cancer patients. Music therapy can help reduce pain and anxiety for cancer patients, including colorectal and gastrointestinal cancer patients. However, more prospective randomized controlled research with large sample sizes is needed to fully understand the effectiveness of music therapy on colorectal cancer patients and to determine the most effective methods for incorporating it into conventional medical treatment plans.

## Figures and Tables

**Figure 1 medicina-59-00482-f001:**
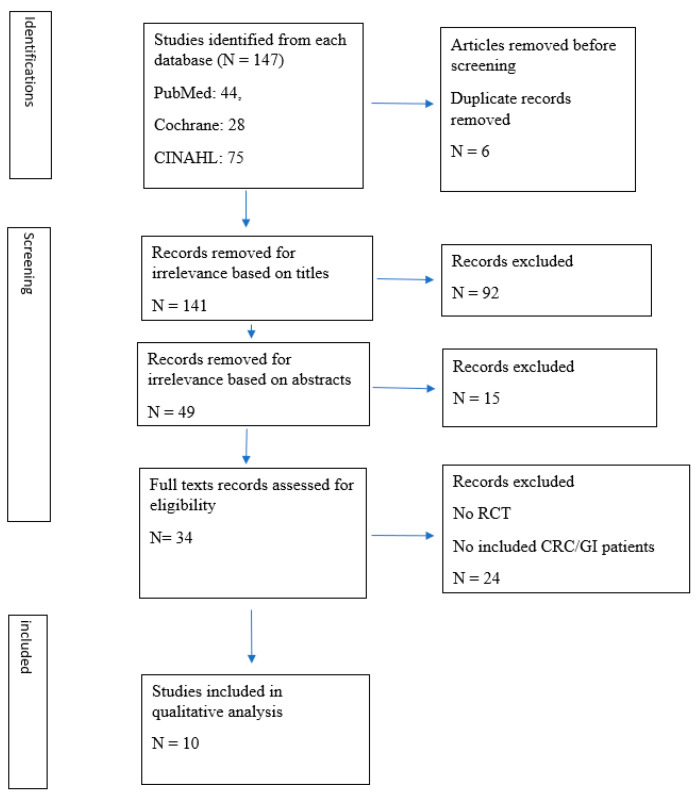
Prisma Flow Chart—Identification of Studies via Databases.

**Table 1 medicina-59-00482-t001:** Summary of Data Extracted from Relevant Studies.

Study	Year	Country	Study Design	Setting	Patients (N)	CRC or GI (%)	MT Type	Treatment	Measure	Main Results
Arruda et al. [31]	2016	Brazil	RCT	In-patients	65	21	Instrumental songs by the artists Yasunori Mitsuda and Vic Mignona	Listened to 30 min clip.	Visual Analog Scale (VAS), the Beck Depression Inventory, the Herth Hope Scale (HHS)	MT improved pain and depression.
Bieligmeyer et al. [32]	2019	Germany	RCT	In-patients	48	14	Vibroacoustic sound bed	A music therapist played for patients who would lay on a sound bed to listen to and feel music.	Basler Mood Questionnaire (BMQ) EORTC-QLQ C30	Vibroacoustic sound bed therapy had greater subjective emotional experiences and well-being.
Fernando et al. [33]	2019	Sri Lanka	RCT	In-patients	24	46	Music from qualified Sri Lankan composers	Listened to a 28 min long music clip.	VAS Wong–Baker Faces Pain Scale	MT sessions significantly reduced pain and anxiety.
Huang et al. [34]	2010	Taiwan	RCT	palliative care and in-patient’s care	126	12	folk songs, Buddhist hymns, harp, and piano	Patients listened to a chosen recording for 30 min.	Sensation and distress of pain VAS scales	MT provided greater pain relief than analgesics alone.
Warth et al. [35]	2021	Germany	RCT	Palliative care	104	29	“Song of Life”, a biographically meaningful and emotionally arousing song, three-session music therapy intervention	Music therapist performed live music with improvisation with a pre- and post-therapy discussion for a total of 30 min.	NCCN distress score	MT reduced distress score and pain.
Düzgün et al. [36]	2021	Turkish	RCT	Palliative care	60	Included but did not report percentage	Turkish classical music	A total of six music sessions, 10 min each with Turkish classical music.	McGill Pain Questionnaire, State-Trait Anxiety Inventory, General Comfort Scale	MT decreased mean pain scores over time, decreased anxiety, and improved comfort level.
Hilliard [37]	2003	USA	RCT	Palliative care/Hospice	80	Included but did not report percentage	Song choice, singing, listening to live music, instrument playing, etc.	MT group was regularly visited by a music therapist for sessions.	Quality of life measures and lifespan	Quality of life was higher in the MT group and continued to improve over time. Length of life had no change.
Hunter et al. [38]	2019	USA	RCT	Patients received chemotherapy	474	6	Soft, relaxation music	Relaxing music with nature sounds or a vocal track.	Morrow Assessment of Nausea and Emesis (MANE)	MT reduced the incidence of mid-chemotherapy.
Kwekkeboom et al. [39]	2003	Brazil	RCT	Outpatients	65	5	Preferred style of music from a variety of music styles	Listened to a CD before and during a painful procedure.	Pain and Speilberger State-Trait Anxiety Inventory	MT showed no significant difference in pain and anxiety.
Clark et al. [40]	2006	USA	RCT	Outpatients	63	9	Preselected or self-selected music	Patients listened to a music therapist with music choices.	Hospital Anxiety and Depression Scale, Distress Numeric Rating Scale, Pain Numeric Rating Scale	MT had a significantly reduced distress and anxiety.

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
