# Peer review of "Music Therapy: A Noninvasive Treatment to Reduce Anxiety and Pain of Colorectal Cancer Patients—A Systemic Literature Review"

_medicina, 2023, doi:10.3390/medicina59030482_

Round 1

Reviewer 1 Report

Thank you for the submission. The authors reviewed about music therapy as a noninvasive treatment to reduce anxiety and pain of colorectal cancer patients.

# Why the one study showed negative results for pain & anxiety reduction?

# Cancer pain may be significantly influenced by cancer stage and metastatic status. In case of other visceral organ or spine/bone metastasis, proper pain management is very hard due to severe pain. The positive effect of music therapy may be different on cancer status. I recommend analyzing correlation pain improvement degree through music therapy and cancer stage.

Author Response

We appreciate the time and effort that you have dedicated to providing your valuable comments on our manuscript. We are grateful for your insightful comments on our paper. We have been able to incorporate changes to reflect most of the suggestions provided by you. We have highlighted the changes within the manuscript.

Your Comment:

Why the one study showed negative results for pain & anxiety reduction?

Response:

The authors included cancer patients who were undergoing noxious medical procedures such as tissue biopsy or vascular port placement in the procedure room of an outpatient oncology clinic.  These procedures were performed by one surgeon. The patients were randomized into different group.

However, the authors acknowledged the limitations and methodological insufficiency which may cause indifference between the groups.

Firstly, sample size is very small: only 24 patients in music group

Secondly, timing of anxiety assessments may not adequate. Anxiety was not measured again until the procedure was completed. but once the procedure started, anxiety may have increased with competing stimuli from the procedure itself.

Thirdly the patient selection bias: in this study, A small number of participants (n = 10) had undergone the same medical procedure at some point in their past. This previous experience could have influenced their experience of pain, anxiety, or perceived control during the current procedure.

Finally, about one-third of patients used medications, the effects of analgesics and anxiolytics may have been sufficient for those participants to be comfortable throughout their procedure.

Your comment:

cancer pain may be significantly influenced by cancer stage and metastatic status. In case of other visceral organ or spine/bone metastasis, proper pain management is very hard due to severe pain. The positive effect of music therapy may be different on cancer status. I recommend analyzing correlation pain improvement degree through music therapy and cancer stage.

Response:

We totally agreed with the reviewer comments. Main limitations with these music therapy studies are small sample size, high heterogeneity in the studies (such as patient population, study setting, outcome measurement methods, music therapy types, and control group selection). Therefore, it is challenging to study the effects of MT by different stage of cancer. more prospective randomized controlled multicenter studies are needed.

In the discussion, we added in the discussion. The effectiveness of music therapy in alleviating pain may differ depending on the cancer stage, further investigation is needed to positive effects of music therapy on cancer patients at different cancer stages

Reviewer 2 Report

Overall, the author presents clearly and concisely. Ask the author to check for typos and consider improving or giving explanations for the following suggestions.

Methods

1. Who is the searcher?

If there are multiple people, do they search independently?

2. Did you conduct your search during August 2022? Please be specific.

In addition, the publication period of the selected articles in this study should also be indicated.

3. There should be additional criteria for exclusion of articles other than duplicity of articles.

Results

4. Although there are variations in the assessment tools. However, the researcher should explain it by possibly categorizing it according to the outcome variables. This may be presented together with the topic of Qualitative measurement and Quantitative measurement.

Author Response

We appreciate the time and effort that you have dedicated to providing your valuable comments on our manuscript. We are grateful for your insightful comments on our paper. We have been able to incorporate changes to reflect most of the suggestions provided by you. We have highlighted the changes within the manuscript.

Your comments:

  1. Who is the searcher?

If there are multiple people, do they search independently?

Response:

We did state in the method:

Two independent researchers (EH, JH) assessed the quality of included studies using the Cochrane risk-of-bias tool for randomized trials.

In acknowledge section: We thank our UCF Librarian Shalu Gillum, JD, MLS, D-AHIP for her contributions to the literature search process.

Your comments:

  1. Did you conduct your search during August 2022? Please be specific.

Response:

We added: The literature search was conducted during August 2022.

Your comments:

In addition, the publication period of the selected articles in this study should also be indicated.

Response:

We added publication period as recommended.

This search was performed by using PubMed/MEDLINE (1966 to Aug 2022), CINAHL (1982 to Aug 2022), and Cochrane Library databases (up to Aug 2022) for all articles meeting search criteria

Your comments:

  1. There should be additional criteria for exclusion of articles other than duplicity of articles.

Response:

We added: Any studies that failed to provide significant relevant data, despite meeting the criteria, were excluded as well

Your comments:

Although there are variations in the assessment tools. However, the researcher should explain it by possibly categorizing it according to the outcome variables. This may be presented together with the topic of Qualitative measurement and Quantitative measurement.

Response:

In the results we did explain each study’s Qualitative measurement and Quantitative measurement:

Huang et al (34) studied cancer patients who were hospitalized in palliative care, oncology, respiratory and gastrointestinal floors. Their study setting was mixed palliative care and in-patient’s care. They proved the music therapy intervention was very helpful by showing their study the effect sizes, d = .64, sensation, and .70, distress. There was significantly less post intervention pain in the music versus the control group, p < .001. Thirty minutes of music therapy created half reduction of pain score in 42% of the music therapy group compared to 8% of the controls. They also demonstrated that the type of music (Taiwanese music or American music) had no impact on the results.

Warth et al (35) showed there was a significant reduction the NCCN distress score in the music therapy group (The effect size -0.51 (-0.86, -0.16)). The NCCN distress score recorded initially as 5, decreased to 3.5 near the end of music therapy. The music therapy group patients also reported higher spiritual well-being (The effect size 0.52 (0.21, 0.84) and ego-integrity (The effect size 0.72 (0.30, 1.13)). However, they found no differences in patients’ functional capacity or their psychological QOL from the control group receiving standard of care treatment.

Düzgün et al (36) showed that music therapy decreased mean pain scores over time (p<0.001), decreased anxiety (p < .001), improved comfortable level (p < .001). However, Düzgün et al did not report the effect size. 

Hilliard (37) found that music therapy conferred significant differences in overall QOL. There are three subscales in the overall QOL: functional, psychophysiological, social/spiritual well-being. The authors did further analysis and found that there were no differences in functional well-being (F (l, 78) = 2.034; p> .05), social, or spiritual well-being (F (l, 78) = 1.278; p > .05) between music therapy group and control group. There was difference in psychophysical well-being (F (l, 78) = 3.995; p< .05).  The authors also showed that music therapy had no impact on patient’s length of life.

Fernando et al (33), and Arruda et al (31)  examined the effects of music therapy on pain, depression, and anxiety. Both studies found that values of pain, anxiety, and depression were significantly diminished. Aurruda et al (31) used Visual Analog Scale (VAS), the Beck Depression Inventory (BDI) to measure pain and depression. They found that music therapy significantly decreased pain (p<0.001) and depression (p = 0.004) (confidence interval = 95%) in patients with cancer who were hospitalized. Additionally, Fernando et al (33) Wong Baker Faces Pain Scale to measure pain, anxiety. They reported significant reductions in pain (P = 0.007, 95% CI: 3.716 to – 1.534), anxiety (P = 0.0022, 95% CI: – 2.465 to – 0.6184), respiratory rate and pupillary size in music therapy group. They found no difference in pulse rate, systolic blood pressure, and diastolic blood pressure between music group and control group. However, this study was crossover study.

Bieligmeyer et al (32) used BMQ (self-rating tool) to assess the current state of mood, and the subscales inner balance, vitality, vigilance, social extroversion. They found music therapy slight increases in balance, vitality, vigilance, and satisfaction as well as significant increases in body warmth, present mood, and overall satisfaction. They did not report the effect size. They found no significant changes in pain levels (P = 0.21).   

Clark et al (40)  measured Hospital Anxiety and Depression Scale (HAD), Distress Numeric Rating Scale, Profile of Mood States (POMS) Fatigue, Pain Numeric Rating Scale. They demonstrated anxiety (P<0.05) and distress (p<0.01) were significantly lower for patients in the music therapy group than the control group. They also found that there were no significant differences between group scores in pain, depression, fatigue. They did not report the effect size.

Kwekkeboom et al (39) studied cancer patients undergoing outpatient medical procedures in outpatient oncology clinic. They measured Pain intensity by using a numeric rating scale, Anxiety by the Speilberger State-Trait Anxiety Inventory–state portion (STAI-s). Pain, post procedure anxiety by patients in the music therapy group did not differ from those in the control group. Pain scores in the music group were X = 2.33 (SD = 0.37), and in control group were X = 1.47 (SD = 0.40). Post procedure anxiety scores were X = 33.45 (SD = 1.77) in the music group, X = 30.59 (SD = 1.93) in the control group.

Round 2

Reviewer 1 Report

Thank you for your revised work. The limitations should be described in the conclusion. Thank you.

Author Response

We appreciate the time and effort that you have dedicated to providing your valuable comments on our manuscript. We are grateful for your insightful comments on our paper. We have been able to incorporate changes to reflect most of the suggestions provided by you. We have highlighted the changes within the manuscript.

Your Comments:

 The limitations should be described in the conclusion.

Response:

 We added this sentence into the conclusion:

In conclusion, this systematic literature review and qualitative analysis supported the efficacy of MT for cancer patients.  Music therapy can help reduce pain and anxiety for cancer patients, including colorectal and gastrointestinal cancer. However, more prospective randomized controlled research with large sample size is needed to fully understand the effectiveness of music therapy on colorectal cancer patients and to determine the most effective methods for incorporating it into conventional medical treatment plans.
